# Serum BDNF Levels in Acute Stroke: A Systematic Review and Meta-Analysis

**DOI:** 10.3390/medicina57030297

**Published:** 2021-03-22

**Authors:** Eleni Karantali, Dimitrios Kazis, Vasileios Papavasileiou, Angeliki Prevezianou, Symela Chatzikonstantinou, Foivos Petridis, Jack McKenna, Alina-Costina Luca, Constantin Trus, Alin Ciobica, Ioannis Mavroudis

**Affiliations:** 1Third Neurological Department, Aristotle University of Thessaloniki, 57010 Thessaloniki, Greece; lena.kar@outlook.com (E.K.); dimitrios.kazis@gmail.com (D.K.); angeliki.prev@gmail.com (A.P.); melina.chatzik@gmail.com (S.C.); f_petridis83@yahoo.gr (F.P.); 2Department of Neurosciences, Leeds Teaching Hospitals NHS Trust, Leeds LS97TF, UK; vpapavasileiou@gmail.com (V.P.); jackscloudinthesky@me.com (J.M.); i.mavroudis@nhs.net (I.M.); 3Faculty of Medicine, “Grigore T. Popa” University of Medicine and Pharmacy, 16 Universitatii Street, 700115 Iasi, Romania; 4Department of Morphological and Functional Sciences, Faculty of Medicine, “Dunărea de Jos” University, 800008 Galați, Romania; 5Department of Biology, Faculty of Biology, Alexandru Ioan Cuza University of Iasi, Carol I Avenue 20A, 700506 Iasi, Romania; alin.ciobica@uaic.ro; 6Center of Biomedical Research, Romanian Academy, 8th Carol I Avenue, 700506 Iasi, Romania; 7Academy of Romanian Scientists, Splaiul Independentei nr. 54, Sector 5, 050094 Bucuresti, Romania

**Keywords:** stroke, BDNF, functional outcome, acute phase, meta-analysis

## Abstract

*Background and objectives:* Brain-derived neurotrophic factor (BDNF) is one of the most studied neurotrophins. Low BDNF concentrations have been noted in patients with traditional cardiovascular disease risk factors and have been associated with the increased risk of stroke/transient ischemic attack (TIA). We aimed to study the correlation of BDNF serum levels with acute stroke severity and its potential role as a biomarker in predicting functional outcome. *Materials and methods:* We systematically searched PubMed, Web of Science, and the Cochrane database using specific keywords. The endpoints examined were the correlation of BDNF with functional outcome, the National Institute of Health stroke scale (NIHSS) measured at the acute phase, and stroke infarct volume. We also compared serum BDNF levels between stroke patients and healthy controls. *Results:* Twenty-six records were included from the initial 3088 identified. Twenty-five studies reported NIHSS and BDNF levels on the first day after acute stroke. Nine studies were further meta-analyzed. A statistically significant negative correlation between NIHSS and BDNF levels during the acute phase of stroke was noted (COR: −0.3013, 95%CI: (−0.4725; −0.1082), z = −3.01, *p* = 0.0026). We also noted that BDNF levels were significantly lower in patients with stroke compared to healthy individuals. Due to the heterogeneity of studies, we only conducted a qualitative analysis regarding serum BDNF and functional outcome, while no correlation between BDNF levels and stroke infarct volume was noted. *Conclusions:* We conclude that in the acute stroke phase, stroke severity is negatively correlated with BDNF levels. Concurrently, patients with acute stroke have significantly lower BDNF levels in serum compared to healthy controls. No correlations between BDNF and stroke infarct volume or functional outcome at follow-up were noted.

## 1. Introduction

Stroke is the second cause of death worldwide and a leading cause of disability [1]. The recent advances in acute ischemic stroke treatment, such as intra-arterial mechanical thrombectomy, improved stroke management and reduced stroke-related disability. Nonetheless, the aging of the population and the increasing prevalence of stroke risk factors are predicted to lead to an increase in stroke survivors. In Europe, the projections forecast a rise of stroke survivors, who will reach the 4,631,050 people in 2035, a 45% increase in stroke-related deaths, and a rise of lost disability-adjusted life years lost (DALYs) by 32% [2]. The need for a potent, non-invasive biomarker which can be used to diagnose and predict the functional outcome of stroke is growing. 

Neurotrophins are a family of various soluble molecules, which are involved in multiple nervous system functions, such as cell growth, differentiation, and plasticity [3]. They act through binding to two classes of transmembrane receptors: the tropomyosin-related tyrosine kinase (Trk) receptor family and the p75NT receptor (p75NTR) [4,5]. In the healthy brain, neurotrophins derive from neurons, while peripheral blood cells and cells of the immune system also produce small amounts [3,6]. Their contribution rises when the neurotrophin levels fall due to a pathological CNS process. Brain-derived neurotrophic factor (BDNF) is one of the most studied neurotrophins, the role of which has been unveiled through animal experiments. In the CNS, mature BDNF mediates in synaptic plasticity, dendritic branching, the regulation of both inhibitory and excitatory neurotransmitters, and neuronal growth, while proBDNF contributes to cell apoptosis [4,7,8]. In healthy individuals, BDNF can be detected in the serum and is mostly stored in platelets. The normal values vary depending on the method used [8]. Female individuals have higher levels of BDNF, while a gradual decrease has been reported with increasing age in both sexes. Several studies show that BDNF levels are activity dependent. Physical activity or cognitive enhancement and social activity enhance BDNF expression. On the other hand, a sedentary lifestyle and obesity can lead to decreased concentrations in BDNF [9].

Alterations of BDNF levels in serum have been reported in epilepsy, various neurodegenerative and psychiatric disorders, such as Alzheimer’s dementia, and schizophrenia [10]. Low concentrations have also been noted in patients with traditional cerebrovascular disease (CVD) risk factors and metabolic syndrome, while a negative correlation between circulating BDNF and body mass index, lipid profile and blood pressure has been observed [11]. In a Framingham sub-study, low BDNF levels in healthy individuals were associated with an increased risk of future stroke/TIA [12]. In acute stroke, low BDNF levels have been correlated with worse scores in the National Institute of Health stroke scale (NIHSS), larger infarct volume, and poor long-term functional outcome [13,14].

We conducted this systematic review and meta-analysis, aiming to study the role of BDNF as a biomarker in predicting the functional outcome of stroke when measured in the acute stroke phase. Concurrently, we tried to define the association between BDNF levels in the acute phase and stroke severity, as expressed with clinical and neuroradiological measures.

## 2. Materials and Methods

We conducted our systematic review and meta-analysis according to the methodological steps of PRISMA guidelines [15]. Using the search strategy: (stroke OR cerebrovascular disease OR CVD) AND (BDNF OR brain-derived neurotrophic factor OR neurotrophin), we systematically searched PubMed, Web of Science Search engines, and the Cochrane database. We imported all the references into a reference manager software [16]. After removing the duplicates, two authors (K.E. and P.A.) independently screened the titles, abstracts, and full texts of the potentially eligible articles. We resolved any arising disagreements through discussion.

### 2.1. Inclusion and Exclusion Criteria

The inclusion criteria were the following: (a) patients with stroke (ischemic or hemorrhagic) (b) BDNF values in serum, (c) acute phase (within 14 days after stroke), (d) participants 17 years old or older, male or female, (e) stroke severity, stroke volume infarct, and functional outcome as the primary or secondary of the study. The studies were excluded according to the following criteria: (a) animal studies, (b) review or meta-analysis, (c) study including below 5 participants, (d) study in healthy individuals.

### 2.2. Study Endpoints

We performed primary efficacy and sensitivity analyses to assess the correlation of serum BDNF levels in the acute stroke phase with functional outcome in patients with acute stroke.

We conducted additional analyses to assess the correlation of BDNF in the acute stroke phase with stroke severity as quantified with the National Institute of Health stroke scale (NIHSS) and infarct volume. We also performed primary efficacy and sensitivity analyses to assess serum BDNF levels in patients with stroke during the acute phase versus healthy controls.

In all cases, if the data could not be further meta-analyzed, we performed a qualitative analysis, by outlining the main findings of each study.

### 2.3. Managing the Missing Data

To perform a comprehensive analysis, we contacted the corresponding authors of each selected article with missing data by email. In cases where additional information was provided, we included the study in our meta-analysis.

### 2.4. Risk of Bias Assessment

We evaluated the risk of bias of the selected articles using the Cochrane risk of bias tool [17]. We rated the studies as low, high, or unclear risk in each of the seven domains examined: random sequence generation, allocation concealment, blinding of participants and personnel, blinding of the outcome, incomplete outcome data, selective reporting, and other sources of bias. Studies with dropouts >10% and without intention-to-treat analysis were evaluated as high risk of bias in the “incomplete outcome data” section.

### 2.5. Statistical Analysis

For the purpose of the statistical analysis and the graphical representation of the data, we used the Meta library in R Studio. The statistical analysis was performed using standardized mean difference methodology, and the medians and interquartile range were converted to means and standard deviation in accordance with the methodology presented by Hozo et al. [18]. We also conducted a correlation analysis between the serum BDNF levels in the acute stroke phase with functional outcome in patients with acute stroke. We used 95% confidence interval and overall heterogeneity was assessed using Cochrane’s Q statistic and I2. Based on the heterogeneity of the data for each comparison, we used a random-effects or fixed-effects model. Publication bias was assessed using Egger’s linear regression test and the visual inspection of funnel plots, and where the distribution was not roughly symmetrical, this was suggestive of an increased risk of bias. For each outcome, we used the inverse variance-weighted average method to calculate the effect of each study on our results, while t2 was measured through the DerSimonian–Laird method.

## 3. Results

### 3.1. Study Selection—Quality Assessment

Our search engine search yielded 3088 studies. The step-by-step screening process is described in Figure 1. Twenty-six studies were selected and further assessed using the Cochrane risk of bias tool. Ten studies were assessed as low (green), one as high (red), and fifteen studies as an unclear (yellow) risk of bias (Figure 2). The basic characteristics of each study are summarized in Table 1.

### 3.2. Functional Outcome at Follow-Up

Twenty-one studies reported both the serum BDNF levels and measured functional outcome on follow up. Despite measuring both BDNF and functional outcome, nine studies did not examine the correlation between them [19,20,21,22,23,24,25,26,27].

Algin et al. correlated BDNF levels measured within 4 h of acute ischemic stroke with stroke mortality [28]. The authors reported that BDNF levels did not differ between stroke survivors (*n* = 65, BDNF = 3.80 ± 1.99 ng/mL) and non-survivors (*n* = 10, BDNF = 4.50 ± 2.39 ng/mL) (*p* = 0.296).

Chan and colleagues studied serum BDNF and the frequency of BDNF positive T cells in patients with acute ischemic stroke and controls [29]. The authors did not measure the correlation between serum BDNF and functional outcome measured with the Barthel index scale (BI) at any time-point. Nonetheless, patients with acute ischemic stroke were found to have an increase in BDNF positive T regulatory cells (Tregs) when compared to controls (*p* < 0.01), measured within 3 weeks after the event. Concurrently, patients with a low percentage of BDNF+ Tregs were found to be more dependent than patients with a high percentage at 6 months. Hutanu et al. examined the role of serum BDNF as a predictor of dependency in daily life activities after an acute ischemic stroke [30]. Plasma BDNF levels measured on admission were lower in patients with lower BI scores (BI ≤ 80) measured at discharge (day 5 post-stroke). An optimal cut-off of 2.33 pg/mL with 91.1% sensitivity, and 31.1% specificity for predicting a poor outcome (BI ≤ 80) was reported. 

Lasek Bal and colleagues studied the role of BDNF in the acute ischemic stroke phase in correlation with functional outcome measured with modified Ranking Scale (mRS) at 90 days [31]. The authors reported that BDNF levels did not vary significantly between the groups with good outcome (mRS = 0–2) and poor outcome (mRS = 3–6).

Mourao et al. studied plasma levels of BDNF in correlation with mRS, NIHSS, and Gugging Swallowing screen scale (GUSS), all measured within the first 24 h after the ischemic stroke, at 72 h, at discharge [32]. Lower BDNF levels at 72 h were correlated with higher NIHSS, higher mRS, and higher difficulty in swallowing measured with GUSS score at the same time-points. Specifically, an adjusted R coefficient of 0.101 was reported (coefficient B = −147.126, *p* = 0.014, 95%CI: −262.940–31.312), leading the authors to conclude that BDNF levels are independently associated with the prognosis of acute stroke patients.

The role of BDNF in the prognosis of patients with hyperglycemia after acute stroke was examined by Otero Ortega et al. [33]. Two groups were formed: patients with post-stroke hyperglycemia (PSHG) and non-PSHG. The authors quantified the functional outcome with mRS at 90 days after an acute ischemic stroke. Blood samples were collected at 24–48 h and 72–96 h after admission. While the unadjusted analysis revealed a significant association of BDNF with poor outcome (mRS = 3–6) at follow-up (3 months after stroke), this finding was not confirmed after the adjusted analysis.

Pedard and colleagues hypothesized that the levels of BDNF containing peripheral blood mononuclear cells (PBMCs) may be connected with functional outcome after acute ischemic stroke [34]. All patients received intravenous thrombolysis (rtPA), and both serum BDNF and BDNF containing PBMCs were measured before and after the intervention. No association was found between serum BDNF and stroke outcome. Nonetheless, the authors noted that PBMCs that contained BDNF measured on the third day after stroke was increased in patients with good functional outcome. In patients with unfavorable outcome, a gradual decrease in BDNF levels in PBMCs before thrombolysis to the third day after stroke was reported (−64%, *p* = 0.013). Finally, after adjusting for other stroke predictors, BDNF-PBMCs predicted functional outcome (OR = 12.4, 95%CI: 1.4–112.2, *p* = 0.046) and the authors reported a cutoff value of 6.66 ng/mg to predict good functional outcome (mRS = 0–2) (sensitivity/specificity = 48.0%/92.9%, respectively).

Sobrino et al. explored the association between various serum biomarkers and functional outcome after ischemic stroke [35]. In this multicenter study, they included 552 patients within the first 24 h after stroke (non-lacunar etiology). Both functional outcome and BDNF levels were measured at admission, day 7, 3 months, and 12 months after stroke. BDNF levels did not differ between patients with a good functional outcome (mRS = 0–2) and those with a poor functional outcome (mRS = 3–6) at any time-point.

Stanne et al. examined the role of serum BDNF as a potential predictor of stroke outcome in patients with acute stroke followed up to seven years after the event [14]. The authors reported that BDNF levels were lowest among patients with poor functional outcome. Although, in regression analysis, BDNF levels were not correlated with the functional outcome on short-term follow-up, at 2 and 7 years after, low levels of BDNF at the acute stroke phase were correlated with poor functional outcome (mRS = 3–6). Similarly, Wang and colleagues tested whether serum BDNF can be served as a stroke outcome predictor or not [36]. The authors reported that BDNF levels measured during the acute stroke phase in patients with poor outcome (mRS = 3–6) were lower compared to those with good outcome (*p* < 0.001) and that BDNF was an independent stroke outcome predictor. A cutoff of 14.5 ng/mL, which yielded an 81.5% sensitivity and 70.5% specificity, predicted a poor outcome (mRS = 3–6). Concurrently, the authors reported that low BDNF levels were an independent predictor of mortality in multivariate logistic regression analysis, BDNF (OR = 4.04, 95%CI: 2.07–9.14), after adjusting for all other significant predictors.

Yang and colleagues tested the role of serum BDNF in predicting depression after stroke [37]. All outcomes were assessed on admission and fourteen days after. Lower BDNF levels on day one post-stroke were correlated with a higher incidence of post-stroke depression (OR = 0.551, 95%CI: 0.389–0.779, *p* = 0.001), with a cut-off value of BDNF < 5.86 ng/mL to be predictive of depression development in the first two weeks. No significant correlation of BDNF measured on day one after stroke and modified BI was noted (Spearman’s Rho = 0.128, *p* = 0.092).

Lastly, Zhang and colleagues examined how treatment with atorvastatin correlated with the functional outcome of stroke and BDNF levels versus the control group [38]. In all patients, linear regression analysis revealed a significant association between BDNF levels and mRS-BI. Specifically, a negative correlation between BDNF and mRS (Spearman’s rho = −0.65, *p* < 0.001) and a positive correlation between BDNF and BI (Spearman’s rho = 0.71, *p* < 0.001) was noted.

### 3.3. NIHSS

Twenty-five studies reported both the serum BDNF levels and the NIHSS during the acute stroke phase. Thirteen studies, despite measuring BDNF and scoring NIHSS, do not examine the in-between correlation [20,21,22,24,25,26,27,33,34,35,39,40,41]. Amongst the rest, nine were further meta-analyzed [14,15,23,28,32,36,37,42,43]. Due to high heterogeneity between the studies (tau2 = 0.0701, H = 3.06 (2.32; 4.05), I2 = 89.3% (81.4%; 93.9%)), we used a random-effects model. In the acute stroke phase, BDNF levels in serum were negatively correlated with NIHSS, which reached statistical significance (random-effects model COR: −0.3013, 95%CI: (−0.4725; −0.1082), z = −3.01, *p* = 0.0026) (Figure 3). For the detection of publication bias, we performed Egger’s regression test and the visual inspection of the funnel plot. As we detected an asymmetry in the funnel plot (t = −0.9399, df = 6, *p* = 0.3836), we additionally performed the rank correlation test for funnel plot asymmetry. The asymmetry detected was not statistically significant; thus, we concluded that the possibility of the publication bias to affect our results is low (Kendall’s tau = 0.2143, *p* = 0.5484) (Figure 3).

We further conducted a sensitivity analysis to assess the effects of each study to our findings. We conclude that our result is robust and is not influenced by a specific study (95%CI: −0.3011 (−0.4487; −0.1374), *p* = 0.0004, t2 = 0.0473, I2 = 89.3%) (Figure 4).

Three additional studies reported the relationship between BDNF and NIHSS during the acute stroke phase. Asadollahi and colleagues studied the effect of the aqueous extract of saffron administration in acute stroke outcome [19]. The authors formed two groups, one treated with standard treatment plus saffron extract and one receiving only standard treatment (statins, antiplatelets, antihypertensive treatment). BDNF was measured in serum on the first and fourth days after stroke. The authors reported a significant increase in BDNF concentration measured at day four post-stroke in the saffron-treated group compared to the control. Concurrently, a negative significant non-linear cubic regression between BDNF and the NIHSS score (cubic regression = 0.573, Sig = 0.0000062) was noted.

Various plasma biomarkers were tested for their efficacy in predicting stroke functional outcome by Hutanu et al. [30]. BDNF in serum collected during the first day of hospitalization was negatively associated with NIHSS measured on discharge (*p* = 0.014). The authors also reported that patients with NIHSS > 7 at discharge had lower BDNF levels than patients with NIHSS ≤ 7.

Lasek Bal and colleagues studied the efficacy of BDNF as a potential predictor of outcome in patients with acute ischemic stroke [31]. Both the NIHSS and BDNF were measured during the first day of stroke. The authors reported that BDNF levels in patients with mild stroke (NIHSS ≤ 4) did not differ significantly compared to patients with NIHSS > 4.

### 3.4. Infarct Volume

Ten studies reported both the serum BDNF levels and the infarct volume. Despite measuring both, three studies did not examine the in-between correlation [20,23,24]. Five studies were further meta-analyzed [15,28,32,36,38]. Due to high heterogeneity between the studies included (tau2 = 0.0326, H = 2.08 (1.33; 3.24), I2 = 76.8% (43.6%; 90.4%)), we used a random-effects model. No statistically significant correlation between serum BDNF and stroke volume was revealed (random-effects model COR: −0.1503, 95%CI: (−0.3286; 0.0383), z = −1.56, *p* = 0.1177) (Figure 5). We further proceeded to an influence analysis, which revealed that omitting one study out at a time has a significant effect on the overall result (95%CI: −0.1503(−0.3286; 0.0383), *p* = 0.1177, t2 = 0.0326, I2 = 76.8%) (Figure 6).

Mourao et al. investigated the association between the Alberta stroke program early CT score (ASPECTS) score and multiple serum biomarkers in patients with acute ischemic stroke [26]. Fifty patients were included (26 with ASPECTS < 10 and 24 with ASPECTS = 10). As expected, a significant negative correlation was reported between ASPECTS and mRS/NIHSS. Concurrently, the authors noted a significant positive correlation between ASPECTS score and BDNF levels (Spearman’s rho = 0.307, *p* = 0.030).

Sobrino et al. examined the correlation between multiple serum biomarkers and ischemic stroke volume measured at various time-points (admission, day 7, 3rd month, 12th month) [35]. No correlation between the two was noted at any time-point (MANOVA test, *p* = 0.081).

### 3.5. BDNF in Stroke Patients versus Controls

A total of seven studies measured serum BDNF and evaluated the differences between stroke patients (six studies included only ischemic stroke patients and one both ischemic and hemorrhagic) and healthy controls. All studies were further meta-analyzed [14,15,25,28,30,36,43]. Due to high heterogeneity between the studies (I2 = 99.49%), we used a random-effects model. A statistically significant difference was demonstrated (Figure 7). Specifically, we found that patients with acute stroke had significantly lower serum BDNF levels compared to healthy controls (Standard mean difference: −2.37 (−4.36, −0.38)). For the detection of publication bias, we performed weighted regression with multiplicative dispersion and visual inspection of the funnel plot (t = −0.1407, df = 6, *p* = 0.8927). We conclude that the possibility of the publication bias to affect our results is low (Figure 7). We further conducted a sensitivity analysis, which confirmed that our result is robust and not affected by a specific study (95%CI: −5.9084(−8.4032; −3.4136), *p* < 0.0001, t2 = 10.5474, I2 = 98.8%) (Figure 8).

## 4. Discussion

Our study confirmed a significant negative association between BDNF and NIHSS; both measured at the acute stroke phase. We also proved that BDNF levels in patients with stroke are significantly increased compared to healthy controls. No association between BDNF and infarct volume was found. Through our qualitative review, we highlighted a potential role of BDNF measured at the acute phase of stroke as a predictor of stroke outcome. We also discussed the findings of individual publications studying the role of BDNF-positive PBMCs and Tregs, in which a correlation with outcome was noted. The limited available data hindered the conduction of a meta-analysis, in which the potential role of BDNF as a predictor of functional outcome would be examined. Thus, a clear-cut association is yet to be revealed.

BDNF has been extensively used in stroke as an indicator of neural regeneration and recovery. Its role in angiogenesis, neurogenesis, brain repair, and synaptic plasticity has been unveiled through animal experiments and has established BDNF as an important component of post-stroke recovery [8,12,44]. MacLellan has also reported a positive correlation between BDNF and post-stroke rehabilitation in stroke models [45]. Nonetheless, the majority of the studies assessing the utility of BDNF derive from animal experiments. Thus, the exact role of BDNF, specifically in serum, in stroke patients is still unclear.

Stanne reported that BDNF levels measured during the acute stroke phase may not be correlated with early functional recovery (3 months post-stroke), but with late functional recovery (2 and 7 years post-stroke) [15]. The authors accredit their findings to the extended period needed for the recovery of various post-stroke deficits (3 to 6 months after stroke) [46]. From the studies we included, the period from stroke onset to follow-up was heterogeneous. The vast majority conducted a follow-up in less than three months, which varied from a few days post-stroke up to 7 years.

We assessed the role of BDNF levels in serum measured during the acute stroke phase. Di Lazzaro previously reported the stability of serum BDNF during the acute stroke phase in ten patients with first-ever acute ischemic stroke [47]. This finding has been hypothesized to be related to an early increase in BDNF in serum after stroke preceding the blood–brain barrier disruption, which follows at 3-4 h after stroke. In the few studies available with consecutive BDNF measurements after stroke, the findings are conflicting. Rodier studied 40 patients, 24 treated with rtPA, and 14 with standard care [42]. BDNF was measured at day 0, 1, 7, and 90 post-stroke. A variation of BDNF levels was indicated, especially in patients not treated with rtPA. Nonetheless, no available correlation between BDNF values was available to assess whether this variation was significant or not.

Concerning the acute stroke severity, from the twenty-three studies examined, seven studies included patients with mild stroke (NIHSS 1–4), while the vast majority of the studies included patients with mild or moderate stroke (NIHSS up to 15). Only one study included patients with moderate to severe stroke and none with severe. Concurrently, increased stroke severity, and even death by stroke were exclusion criteria in some of the studies included, especially in some assessing for depression as a primary endpoint. We conclude that the data regarding the utility of BDNF in patients with severe infarcts has not been extensively studied. Nonetheless, both stroke severity, the timing of BDNF measurement after stroke, and all the other study characteristics reported were comparable among the studies we included (Table 1).

We conducted a thorough review of all the available data concerning the relationship between BDNF, stroke outcome, and acute stroke severity. To our knowledge, this is the first time a negative correlation between acute stroke severity and BDNF and a significant decrease in BDNF levels in patients with stroke compared to controls has been reported. Our meta-analysis has some limitations. First, the differences in methodology among the included studies may have influenced our results. Nonetheless, the heterogeneity did not modify the significance of the effect in the meta-regression and the results of our sensitivity analyses were robust in most cases. Second, the findings of the studies included cannot be generalized to the population, as they derive from single-center studies. From the initial 26 studies included in our systematic review, only 11 studies were included in our meta-analysis due to insufficient data available. Finally, we restricted our search strategy in studies published in the English language.

Future studies should focus on defining the role of serum BDNF in predicting functional outcome measures from early phase up to 3–6 months after stroke. Regarding the role of BDNF as an acute stroke biomarker, the data available are insufficient, and further research is needed.

## Figures and Tables

**Figure 1 medicina-57-00297-f001:**
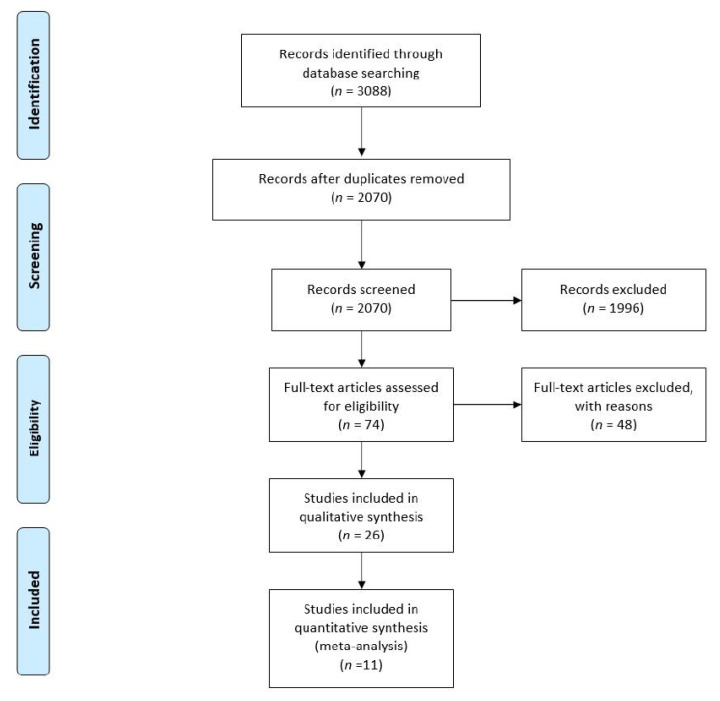
Prisma flow chart.

**Figure 2 medicina-57-00297-f002:**
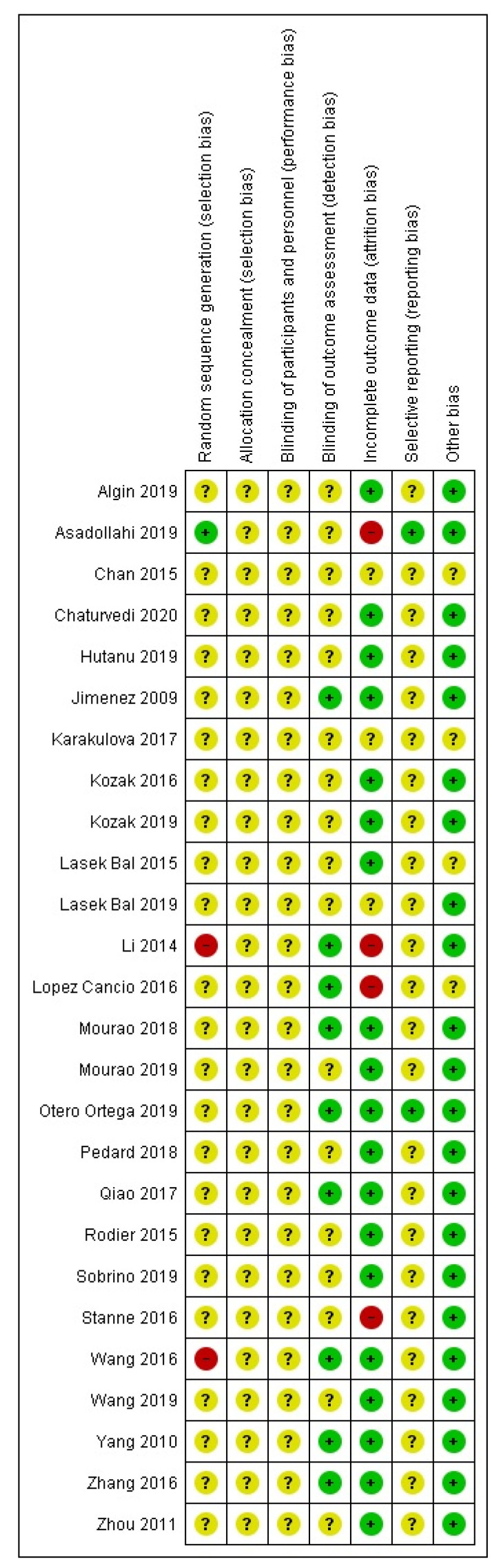
Risk of bias assessment.

**Figure 3 medicina-57-00297-f003:**
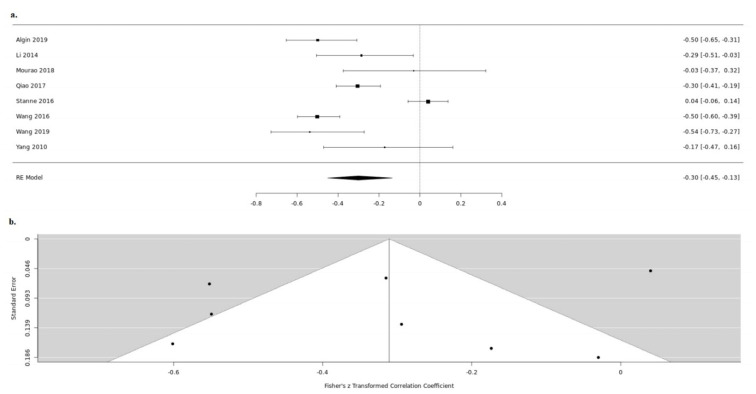
Serum BDNF and NIHSS up (**a**). forest and down (**b**). funnel plot.

**Figure 4 medicina-57-00297-f004:**
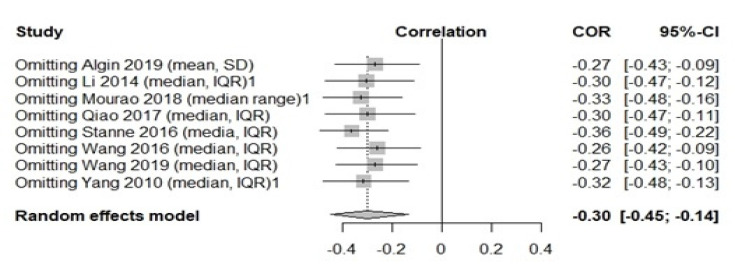
Sensitivity analysis serum BDNF and NIHSS.

**Figure 5 medicina-57-00297-f005:**
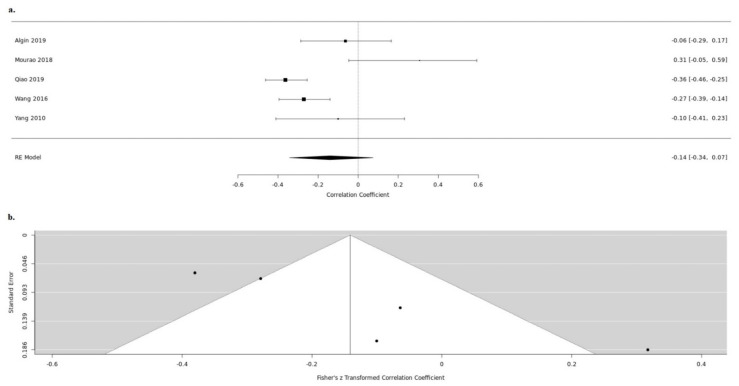
Serum BDNF and infarct stroke volume up (**a**). forest and down (**b**). funnel plot.

**Figure 6 medicina-57-00297-f006:**
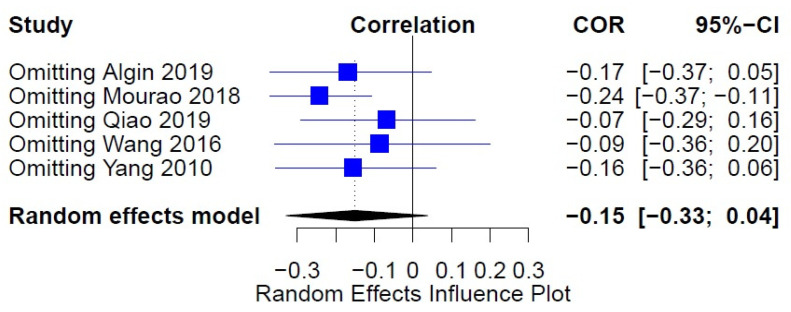
Sensitivity analysis serum BDNF and infarct stroke volume.

**Figure 7 medicina-57-00297-f007:**
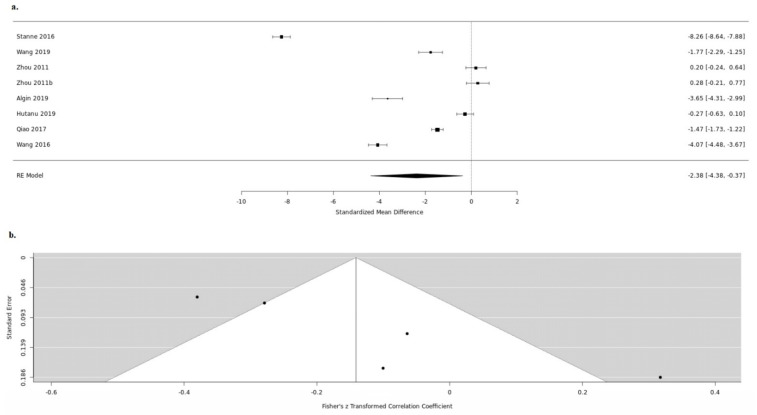
Serum BDNF in stroke patients versus controls up (**a**). forest and down (**b**). funnel plot.

**Figure 8 medicina-57-00297-f008:**
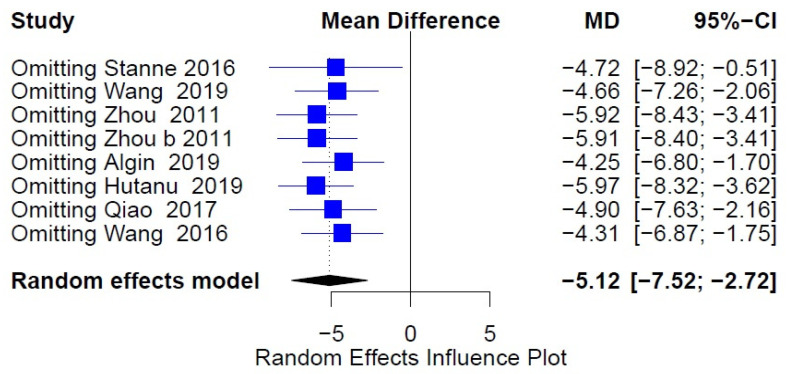
Sensitivity analysis serum BDNF in stroke patients versus controls.

**Table 1 medicina-57-00297-t001:** Study characteristics.

Study	Sample	Type of Stroke	Assay	Time of Blood Collection	No of Patients	NIHSS Baseline	BDNF (ng/mL)	Age (Years)	Outcome Assessment	Time of Follow-Up	Patients Who Received rtPA
**Algin 2019**	Serum	Ischemic stroke	ELISA	Within 4 h after stroke	75	10.88 ± 7.72	3.89 ± 2.05	73.22 ± 11.57	NA	No follow-up	NA
**Asadollahi 2019**	Serum	Ischemic stroke	ELISA	1st and 4th day after stroke	19 saffron- treated	11.35 (9.3–13.3)	1.81 (1.1–2.5)	70.16 ± 11.5	BI	3 months	None
					20 routine care	11.26 (9.4–13)	1.99 (1.2–2.7)	72.25 ± 10			
**Chan 2015**	Serum	Ischemic stroke	ELISA	Day 1, week 1 and 3	75	3.5 (0–24)	NA	69 (27–89)	mRS	6 months	6
**Chaturvedi 2020**	Serum	Any type	ELISA	Admission, week 2, 6th month	104 diabetic	9.36 ± 4.17	11.08 ± 3.85	56.29 ± 11.06	FIM	3 months	NA
					104 non-diabetic	8.01 ± 4.14	8.77 ± 4.04	57.42 ± 10.35			
**Hutanu 2019**	Serum	Ischemic stroke	Xmap	Day 1and 5	114	NA	4.1 (2.73, 9.24)	71.7 ± 10.2	BI, mRS	No follow-up	none
**Jimenez 2009**	Serum	Ischemic stroke	ELISA	Day 7 ± 2 and 30 ± 7 after stroke	25 PSD	3 (1–7)	13.6 (9.8–20.1)	76.6 ± 7.8	BI, mRS	1 month	NA
					109 non-PSD	2 (1–5)	12.9 (10.6–16.1)	69.5 ± 9.6			
**Karakulova 2017**	Serum	Ischemic stroke	ELISA	Days after stroke not mentioned	25 Cytoflavin	6.70 ± 0.62	0.648 ± 0.095	52–74 years old	BI	2 months	none
					27 routine care	6.32 ± 0.26	0.598 ± 0.180				
**Kozak 2016**	Serum	Ischemic stroke	ELISA	On the 1st day of stroke	11 with delirium	10.36 ± 5.88	0.906 ± 0.654	64.63 ± 13.18	NA	No follow-up	none
					49 without delirium	6.10 ± 3.80	1.035 ± 0.761	72.91 ± 5.61			
**Kozak 2019**	Serum	Ischemic stroke	ELISA	On the 1st day of stroke	17 PSD	6.96 ± 4.75	0.751 ± 0.643	67.47 ± 11.20	NA	No follow-up	none
					36 non-PSD		0.729 ± 0.501	65.25 ± 13.98			
**Lasek-Bal 2015**	Serum	Ischemic stroke	ELISA	On the 1st day of stroke	87	NA	9.96 ± 5.21	71.7 ± 11.8	mRS	90 days	none
**Lasek-Bal 2019**	Serum	Ischemic stroke	ELISA	On the 1st day of stroke	138	3 [1,2,3,4,5,6,7,8,9,10,11,12,13,14,15,16,17,18]	NA	73.11 ± 11.48	mRS	30days	53
**Li 2014**	Serum	Ischemic stroke	ELISA	On the 1st day of stroke	59 PSD	8 (4–14)	8.1 (5.6–9.4)	72.8 (11.2)	mRS	3 months	NA
					157 non-PSD	5 (2–8)	13.7 (10.4–16.5)	63.6 (9.1)			
**Lopez Cancio 2016**	Serum	Ischemic stroke	ELISA	Day 1, 7 and 3 months	83	17 [12,13,14,15,16,17,18,19,20,21]	3.3 ± 0.9	69.6 ± 10.9	mRS	3 months	NA
**Mourao 2018**	Serum	Ischemic stroke	ELISA	Up to 24 h, 72 h and discharge	32 (length of stay ≤ 10 days)	5.6 ± 3.8	9.715 ± 2.56	65.5 ± 11.7	mRS	No follow-up	NA
					18 (length of stay > 10 days)	12.4 ± 7.4	9.737 ± 2.87				
**Mourao 2019**	Serum	Ischemic stroke	ELISA	Admission	26 ASPECTS < 10	10.5 ± 6.4	9.05 ± 2.29	67.1 ± 12	mRS	No follow-up	None
					24 ASPECTS ≥ 10	5.8 ± 4.9	10.45 ± 2.86	63.8 ± 11.4			
**Otero Ortega 2019**	Serum	Ischemic stroke	ELISA	At 24–48 h and 72–96 h	95 Non-PSHG	5 (3–11)	12.3 (6.5)	69.7 (12.5)	mRS	72–96 h	NA
					79 PSHG	4 (2–8)	11.6 (6.6)	72.1 (9.3)			
**Pedard 2018**	Serum and PBMC	Ischemic stroke	ELISA	Before fibrinolysis—day 1 and 3	25 mRS 0–2	7 (4–9)	30.4 (27.8–34.0)	76 (65.8–83.5)	mRS	No follow-up	All
					15 mRS 3–6	8 (5–12)	31.3 (24.6–34.8)	81 (70–88)			
**Qiao 2017**	Serum	Ischemic stroke	ELISA	On 1st day of stroke	270	7 (4–12)	22.1 (14.5–27.5)	65 (56–73)	NA	No follow-up	58
**Rodier 2015**	Serum	Ischemic stroke	ELISA	Day 0, 1, 7 and 90	14 non rtPA treated	11.44 ± 2.25	NA	74.71 ± 3.55	NA	90 days	24
					24 rtPA treated	11.20 ± 1.34		69.13 ± 3.01			
**Sobrino 2019**	Serum	Ischemic stroke	ELISA	Admission, 3rd months and 12 months	351 good outcome	8 (5, 13)		66.9 ± 11.6	mRS	12 months	198
					201 poor outcome	18 (14, 20)		69.6 ± 9.4			
**Stanne 2016**	Serum	Ischemic stroke	ELISA	Within 10 days after stroke	491	3 (2, 7)	15.4 ± 5.9	58 (51–63)	mRS	3months, 2 and 7 years	NA
**Wang 2016**	Serum	Ischemic stroke	ELISA	On 1st day of admission (until 48 h)	204	6 (3–12)	13.4 (9.2–16.9)	64 (55–75)	mRS	3 months	41
**Wang 2019**	Serum	Ischemic stroke	ELISA	NA	40	10 (5, 15)	19.14 ± 4.87	63 (50–75)	NA	No follow-up	NA
**Yang 2010**	Serum	Ischemic stroke	ELISA	24–48 h after stroke	37 PSD	7 (4–9.5)	NA	68.95 ± 9.28	mBI	14 days	None
					63 non-PSD	3 (2–4)		68.43 ± 11.18			
**Zhang 2016**	Serum	Ischemic stroke	ELISA	On the 1st day of stroke	37 received statin	7.87 ± 2.26	32.95 ± 6.14	65.11 ± 10.72	mRS, BI	6 weeks	NA
					38 not received statin	8.36 ± 2.93	23.06 ± 5.13	63.34 ± 9.67			
**Zhou 2011**	Serum	Any type	ELISA	Within 7 days after stroke	35 PSD	7 (1, 24)	29.1 ± 11.4	61.7 ± 8.5	mRS, BI	6 months	NA
					58 non-PSD	5 (1, 13)	28.1 ± 9.7	63.5 ± 12.5			

Abbreviations: ASPECTS: Alberta stroke program early CT score; BDNF: brain-derived neurotrophic factor; BI: Barthel index; ELISA: enzyme-linked immunosorbent assay; FIM: functional independence measure; mRS: modified Rankin scale; NA: not available; NIHSS: National Institute of Health stroke scale; PSD: post-stroke depression; PSHG: post-stroke hyperglycemia; rtPA: recombinant tissue plasminogen activator.

## Data Availability

All the current data is available on request from the authors.

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
