# Peer review of "Serum BDNF Levels in Acute Stroke: A Systematic Review and Meta-Analysis"

_medicina, 2021, doi:10.3390/medicina57030297_

Round 1
Reviewer 1 Report
An interesting study, carried out correctly, although limited by a small number of compared studies, finally in the meta-analysis. The observed effect seems to exist despite the heteroenicity of the individual results. At this stage of the research, generalization to the population is not appropriate yet. The relationship between BDNF serum levels and acute stroke severity is important if it could be used to create a predictive marker model. This requires further research, but this work is the first step in that direction. I appreciate the work done in the study,
Author Response
Reviewer 1#
An interesting study, carried out correctly, although limited by a small number of compared studies, finally in the meta-analysis. The observed effect seems to exist despite the heteroenicity of the individual results. At this stage of the research, generalization to the population is not appropriate yet. The relationship between BDNF serum levels and acute stroke severity is important if it could be used to create a predictive marker model. This requires further research, but this work is the first step in that direction. I appreciate the work done in the study
Response: Thank you for your kind and positive comments
Reviewer 2 Report
Thank you for asking me to review the manuscript titled, “Serum BDNF levels in acute stroke: a systematic review and meta-analysis.” The following are my comments, section by section.
Title
- The title is appropriate.
Abstract
- The abstract is very well written.
- Please spell out TIA and NIHSS to be clear
- Otherwise, I have no other comments for this section is it is concise and clear.
Introduction
- Please provide a reference for stroke being the “second most frequent cause of death worldwide…”, and also, please word the sentence as such.
- Please spell out DALYs (line 48)
- The Introduction is good but lacks mention of how BDNF levels are related to stroke. This could be inserted after the paragraph ending on line 72, p.2.
Materials and Methods
- Please provide a reference for the PRISMA guidelines (line 79)
- Line 87 - how was the acute phase of stroke defined?
- Line 99 - can you briefly outline what the qualitative analysis consisted of?
- Line 105 - please provide a reference for the Cochrane Risk of Bias tool
Results
- For Figure 1, please define what each colored dot means as it was not immediately clear to me at first glance.
- Figures 3, 5, and 7 are difficult to read and should be enlarged prior to publication
- The reviews of studies are clear, concise, and useful for readers looking for critical information quickly.
- The statistical methodology used appears to be appropriate, though I am not an expert in this area.
- Overall, the results section is clear and well organized.
Discussion
- The discussion is appropriate.
- The limitations are appropriate.
Overall, the manuscript provides clear data on the relationship between BDNF and stroke outcomes/severity across multiple studies. I congratulate the authors on their manuscript and wish them the best with their future endeavors. I suggest minor revisions as outlined above, most of which are editorial in nature. Thank you again for asking me to review this manuscript.
Author Response
Reviewer 2#
Title
The title is appropriate.
Response: Thank you very much
Abstract
The abstract is very well written.
Please spell out TIA and NIHSS to be clear
Otherwise, I have no other comments for this section is it is concise and clear.
Response: Thank you for your feedback. We spelled out the aforementioned abbreviations.
Introduction
Please provide a reference for stroke being the “second most frequent cause of death worldwide…”, and also, please word the sentence as such.
Response: We included the reference: “1. WHO, Global Health Estimates. “The Top 10 Causes of Death.” World Health Organization, 9 Dec. 2020, www.who.int/news-room/fact-sheets/detail/the-top-10-causes-of-death.”
Please spell out DALYs (line 48)
Response: Thank you for your comment. We spelled out the aforementioned abbreviation.
The Introduction is good but lacks mention of how BDNF levels are related to stroke. This could be inserted after the paragraph ending on line 72, p.2.
Response: Thank you for your feedback. We included the following sentence: In acute stroke, low BDNF levels have been correlated with worse score in National Institute of Health Stroke Scale (NIHSS), larger infarct volume, and poor long-term functional outcome [13, 14].
Materials and Methods
Please provide a reference for the PRISMA guidelines (line 79)
Response: We included the reference: “15. Moher D, Liberati A, Tetzlaff J, Altman DG, The PRISMA Group (2009). Preferred Report-ing Items for Systematic Reviews and Meta-Analyses: The PRISMA Statement. PLoS Med 6(7): e1000097. doi:10.1371/journal.pmed1000097”
Line 87 - how was the acute phase of stroke defined?
Response: Thank you for your comment. We included the acute stroke phase definition in the text. Specifically : ” The inclusion criteria were the following: a. Patients with stroke (ischemic or hemorrhagic) b. BDNF values in serum, c. Acute phase of stroke (within 14 days after stroke),”
Line 99 - can you briefly outline what the qualitative analysis consisted of?
Response: Thank you for your feedback. We reformed our sentence into : “In all cases, if the data could not be further meta-analyzed, we performed a qualitative analysis, by outlining the main findings of each study.”
Line 105 - please provide a reference for the Cochrane Risk of Bias tool
Response: We included the reference: “17. Higgins, J. P., & Green, S. (2008). Cochrane Handbook for Systematic Reviews. The Cochrane Collaboration. https://doi.org/10.1002/9780470712184”
Results
For Figure 1, please define what each colored dot means as it was not immediately clear to me at first glance.
Response: Thank you for your comment. We define the meaning of each color. Specifically, “Ten studies were assessed as low (green), one as high (red), and fifteen studies as unclear (yellow) risk of bias”
Figures 3, 5, and 7 are difficult to read and should be enlarged prior to publication
Response: Thank you for your comment. We attached the figures in the raw form now.
The reviews of studies are clear, concise, and useful for readers looking for critical information quickly.
The statistical methodology used appears to be appropriate, though I am not an expert in this area.
Overall, the results section is clear and well organized.
Response: Thank you for your kind words.
Discussion
The discussion is appropriate.
The limitations are appropriate.
Overall, the manuscript provides clear data on the relationship between BDNF and stroke outcomes/severity across multiple studies. I congratulate the authors on their manuscript and wish them the best with their future endeavors. I suggest minor revisions as outlined above, most of which are editorial in nature. Thank you again for asking me to review this manuscript.
Response: Thank you for your positive comments!